# Lymph node metastasis prediction of papillary thyroid carcinoma based on transfer learning radiomics

Jinhua Yu [1,2], Yinhui Deng [1,3], Tongtong Liu[1], Jin Zhou[4], Xiaohong Jia[5], Tianlei Xiao[1], Shichong Zhou[4], Jiawei Li[4], Yi Guo[1], Yuanyuan Wang [1,2 ✉], Jianqiao Zhou [5 ✉] & Cai Chang [4 ✉]

Non-invasive assessment of the risk of lymph node metastasis (LNM) in patients with papillary thyroid carcinoma (PTC) is of great value for the treatment option selection. The purpose of this paper is to develop a transfer learning radiomics (TLR) model for preoperative prediction of LNM in PTC patients in a multicenter, cross-machine, multi-operator scenario. Here we report the TLR model produces a stable LNM prediction. In the experiments of cross-validation and independent testing of the main cohort according to diagnostic time, machine, and operator, the TLR achieves an average area under the curve (AUC) of 0.90. In the other two independent cohorts, TLR also achieves 0.93 AUC, and this performance is statistically better than the other three methods according to Delong test. Decision curve analysis also proves that the TLR model brings more benefit to PTC patients than other methods.

[1] Department of Electronic Engineering, Fudan University, Shanghai, China. [2] Key Laboratory of Medical Imaging Computing and Computer Assisted Intervention, Shanghai, China. [3] MingGe Research, Fudan University Science Park, Shanghai, China. [4] Fudan University Shanghai Cancer Center, Shanghai, China. [5] Ruijin Hospital Affiliated to Shanghai Jiaotong University, Shanghai, China. ✉email: yywang@fudan.edu.cn; zhousu30@126.com; changc61@163.com

According to the cancer statistics[1], thyroid cancer causes 567,000 cases worldwide, and the incidence rate is ranked ninth. Global incidence ratio of women is 10.2 per 100,000 people, which is three times that of men. The incidence of thyroid cancer has continued to increase in many countries since the 1980s. This is mainly due to the increase of the papillary thyroid carcinoma (PTC) detection rate through the improvement in detection and diagnosis[2].

About 84% of patients with thyroid cancer are PTC, which is the most common thyroid malignancy[3,4]. Between 1974 and 2013, the average incidence rate of PTC was about 6.66%, the incidence-based mortality 0.20% in the United States of America[3]. Although PTC is regarded as an indolent tumor, a portion of cancer cells will metastasize to lymph nodes around the thyroid gland[5], mainly including central lymph node metastasis (LNM) and lateral cervical LNM[6]. Usually, LNM occurs first in the central region, followed by the lateral region. LNM is an important indicator of PTC prognosis, scope and way of surgery, and is also an important risk factor for high recurrence rate and low survival rate of patients[7,8].

With the continuous progress of diagnostic techniques, especially the resolution improvement of the ultrasonography, the detection rate of small thyroid nodules keeps increasing[1]. Overdiagnosis of PTC has become a global consensus[9]. The potential risk of LNM of PTC has led to a large number of PTC patients received treatment such as total thyroidectomy and lymph node dissection (LND), resulting in widespread overtreatment[10]. Ultrasound assessment of cervical lymph nodes is recommended for all patients with confirmed or suspected thyroid nodules[5]. Preoperative ultrasound is a valuable tool in assessing lateral cervical LNM in patients with PTC and can provide relatively reliable information of the lateral neck to assist in surgical management[11]. However, the identification of central cervical LNM by ultrasound has encountered significant challenges. Preoperative ultrasound can only detect 20–31% of central cervical LNM, and may only change the surgical procedure of 20% patients[12–14]. There is an urgent need for a nondestructive and efficient method for predicting the risk of LNM in PTC patients and guiding the clinical diagnosis and treatment process.

Several studies have been proposed for the LNM risk assessment of PTC patients. Some manifested that tumor size, tumor location, tumor extension, microcalcifications, and Hashimoto's disease are independent risk factor of LNM[9,15–18]. Some combined the above risk factors with blood markers such as thyroid stimulating hormone (TSH) and antithyroglobulin antibodies (TGAb) to construct LNM prediction models for PTC patients[19,20]. In recent years, radiomics has attracted much attention in the precise diagnosis. Radiomics-based methods were also proposed for LNM prediction in PTC patients by converting ultrasound images into mineable data[21,22]. These methods extracted features such as intensity, boundary, texture, and wavelet from the ultrasound images, and established the relationship between these high-throughput features and LNM status. Among the above studies, whether based on clinical statistics or radiomics, since the completeness of the extracted image features is difficult to guarantee, the LNM prediction performance was not ideal with the area under the receiver operating characteristic (ROC) curve (AUC) on independent testing set approximately ranged from 0.67 to 0.78.

In this study, we establish a transfer learning radiomics (TLR) model based on B-mode ultrasound images of thyroid lesions to predict LNM risk of PTC patients. In multicenter, cross-machine, multi-operator scenario, we comprehensively compare the diagnostic performance of clinical statistical model (SM), traditional radiomics model (RM), nontransfer learning model, and TLR model. The results show that the TLR achieve stable LNM prediction performance when the data collection protocol is inconsistent, which is close to the actual clinical diagnosis scenario.

## Results

**Main cohort**. We first divided the main cohort into cross-validation set and testing set in order of diagnosis time with the ratio of 8:2. Figure 1 illustrated the ROC curves of the LNM prediction results of the four models on the main cohort. From the experimental results, the AUCs of the four models on the testing set were 0.83, 0.64, 0.82, and 0.93, respectively. The corresponding quantitative indexes of four models were summarized in Table 1.

In the main cohort, three types of ultrasound machines (GE, SuperSonic, and Kretztechnik) collected the most data. In order to verify the impact of different ultrasound machines on the prediction results, we extracted the data of these three types of machines as testing sets and data of other machines as training sets in turn.

In addition, in order to verify the operator's impact on the LNM prediction results, we used the data collected by the three sonographers, who contributed the most in the main cohort collection, as independent testing sets, while data collected by other 21 sonographers as training sets.

When using the data from three machines and three doctors as the independent testing sets, the average ROC values of TLR was 0.887. The ROC curves of the TLR model were shown in Fig. 2, and the corresponding quantitative indexes were summarized in Supplementary Tables 1 and 2, respectively.

**Independent testing cohort**. In independent testing set 1, ultrasound image of each nodule of the multifocal-lesion case is sent to the transfer learning model, and once a nodule is predicted by the model as LNM, the case is classified as LNM positive. Another 513 PTC cases acquired from different hospitals were used independent testing set 2.

The ROC curves and decision curves of the four models in the two independent testing set were compared in Fig. 3. The quantitative index comparisons of the four models and the Delong test results on the three cohorts were summarized in Table 2.

Figure 4 showed the ultrasound images of 20 cases without or with LNM, respectively, and the visualization of their corresponding network features. It can be seen that there was no consistent and significant differences between the case of non-LNM and LNM in terms of the characteristics of the original ultrasound images. However, when the original images passed the transfer learning network, the network features showed obvious differences.

We also compared the proposed TLR model with VGG, ResNet, and Inception ResNet in the three cohorts. The quantitative index comparisons results on the three cohorts were summarized in Supplementary Table 3.

To further illustrate the effect of transfer learning strategy and hyperparameter optimization based on simulated annealing algorithm on the TLR model, an ablation experiment was carried out. For the convenience of description, we denote the TLR with or without the transfer learning as T+ and T−, and the TLR with or without hyperparameter optimization as H+ and H−. Supplementary Table 4 gives the experimental results.

The used graphics card is TITAN XP with the CUDA core number as 3840 and the graphic memory as 45008 MB. For the coding of the deep model, the applied TensorFlow is the GPU version of 1.14.0 and the Keras is utilized with its 2.3.0 version. For the establishment of one deep model in our study, the model

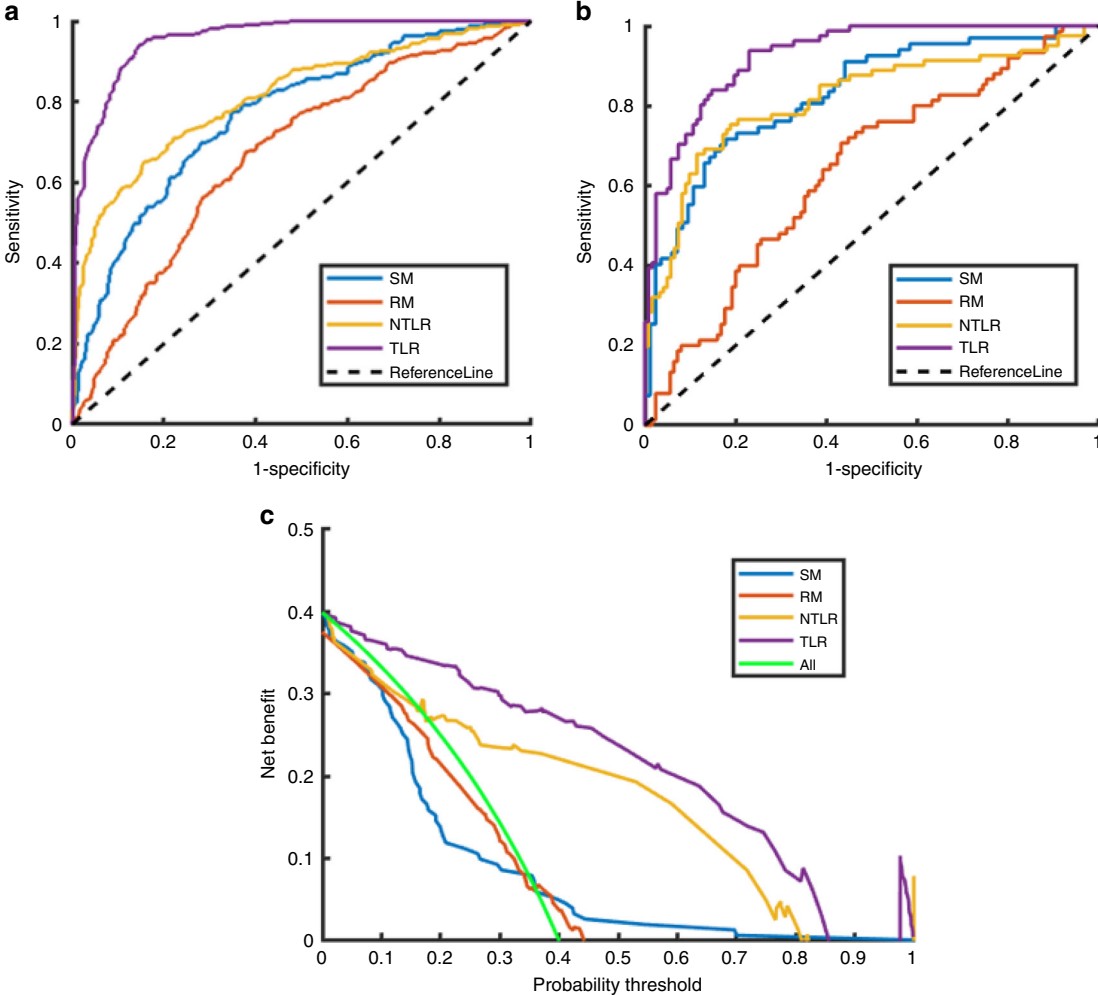

**Fig. 1 Comparison of ROC curves of LNM prediction in the main cohort by four models. a** ROC curves in the cross-validation set, **b** ROC in testing set, and **c** decision curves in the testing set. (SM statistical model, RM traditional radiomics model, NTLR nontransfer learning radiomics, TLR transfer learning radiomics).

**Table 1 Quantitative indexes of four models on main cohort data.**

| Method | | AUC | ACC | SENS | SPEC | PPV | NPV | MCC | F1 score |
|---|---|---|---|---|---|---|---|---|---|
| SM | Cross-validation | 0.77 | 0.70 | 0.77 | 0.65 | 0.60 | 0.81 | 0.42 | 0.68 |
| | Testing | 0.83 | 0.77 | 0.72 | 0.82 | 0.76 | 0.78 | 0.54 | 0.74 |
| RM | Cross-validation | 0.67 | 0.64 | 0.68 | 0.62 | 0.55 | 0.74 | 0.29 | 0.61 |
| | Testing | 0.64 | 0.62 | 0.71 | 0.57 | 0.50 | 0.76 | 0.27 | 0.58 |
| NTLR | Cross-validation | 0.81 | 0.76 | 0.64 | 0.85 | 0.73 | 0.78 | 0.50 | 0.69 |
| | Testing | 0.82 | 0.79 | 0.75 | 0.81 | 0.73 | 0.83 | 0.56 | 0.74 |
| TLR | Cross-validation | 0.96 | 0.89 | 0.94 | 0.85 | 0.81 | 0.95 | 0.78 | 0.87 |
| | Testing | 0.93 | 0.84 | 0.94 | 0.77 | 0.73 | 0.95 | 0.69 | 0.82 |

training process generally takes 6 days with the utilization of hyperparametric optimization. For the prediction on one image data, the time cost of the model inference is around 10 ms.

## Discussion

Although PTC is generally an indolent tumor, LNM will occur in an early stage. The most common site of LNM from the PTC is the central compartment of the neck. The decision to perform central LND during a thyroidectomy usually depends on whether a lymph node suspected of being malignancy can be identified preoperatively. If the node is known to spread to the central neck node, the consensus is to delete all nodes in the area. However,

the utility of central LND for prophylactic reasons remains significant controversial[23]. Undoubtedly, unnecessary cleaning of the central lymph nodes will lead to more injury of recurrent laryngeal nerve. LND of the lateral neck is involved in the expansion of the surgery scope, the injury of the spinal accessory nerve or vagus nerve, and the leakage of the chyle. Therefore, LND is performed only in patients who are diagnosed as LNM positive before surgery.

Ultrasound is an important method for lymph node detection. It is recommended to perform cervical lymph node ultrasound evaluation on all patients with confirmed or suspected thyroid nodules[5]. It is obvious that ultrasound is particularly important

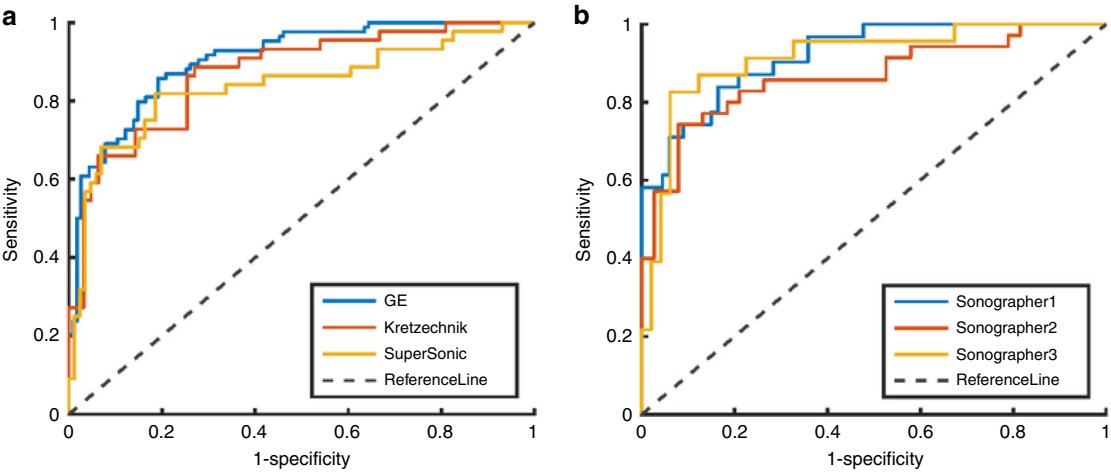

**Fig. 2 Comparison of ROC curves when data from different machines and different doctors are used as independent test sets. a** shows ROC curves when data from different machines, **b** from different doctors.

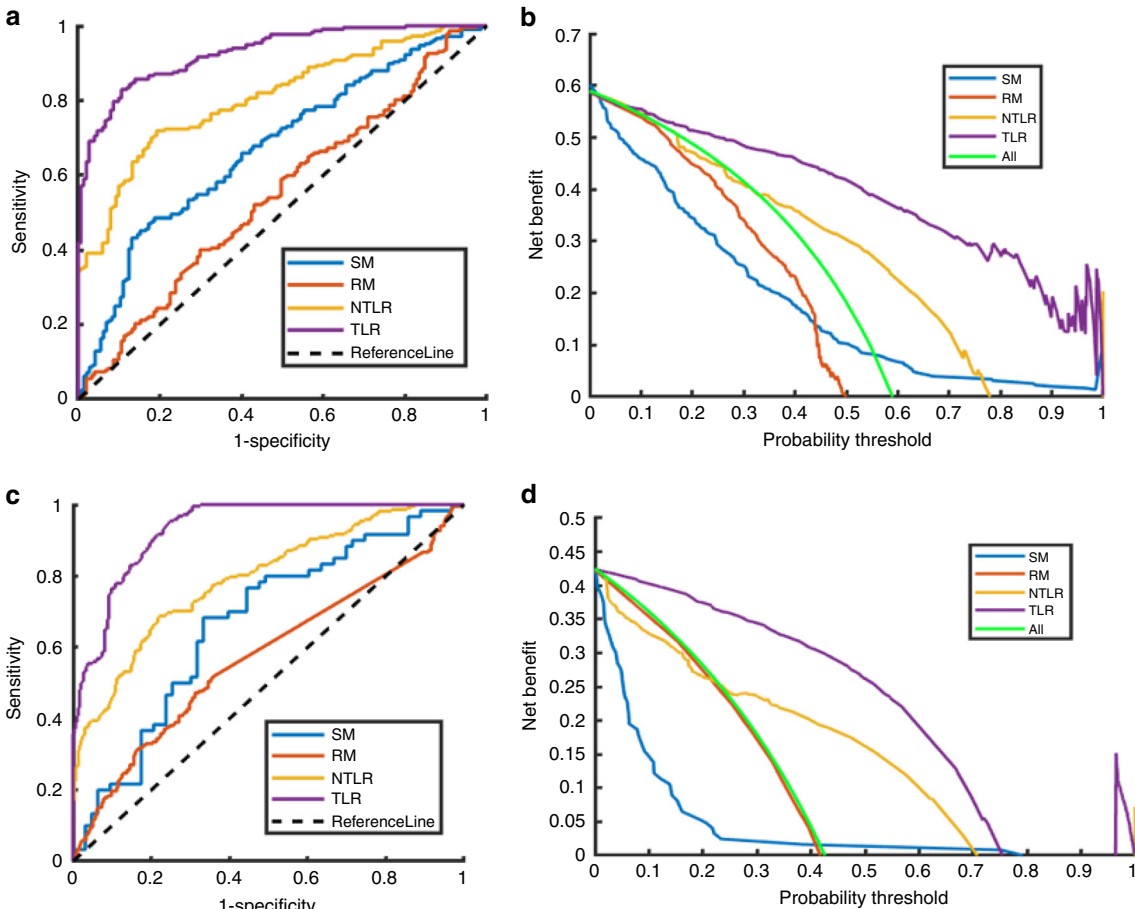

**Fig. 3 Comparison of ROC curves and decision curves of LNM prediction in the two independent testing cohort by four models. a**, **c** ROC curves in the independent testing set 1 and 2, respectively. **b**, **d** Decision curves in the independent testing set 1 and 2, respectively. (SM statistical model, RM traditional radiomics model, NTLR nontransfer learning radiomics, TLR transfer learning radiomics).

for the LNM risk evaluation for PTC patients. LNM in the lateral cervical region often have typical ultrasound features, including microcalcifications, partially cystic appearance, increased vascularization, and hyperechogenicity[5]. However, LNM in the central region lacked these typical ultrasound features. Moreover, there are anatomic areas of the central region that are not well visualized by ultrasound, such as the posterior tracheal area, posterior

esophageal area, posterior pharyngeal area, and mediastinal area. Thus, ultrasound has a relatively high accuracy in the detection and diagnosis of LNM in the lateral region, In the identification of central cervical LNM, ultrasound has encountered great challenges[24].

Furthermore, the diagnostic accuracy of ultrasound on LNM is severely affected by operator differences. Therefore, we urgently

**Table 2 Quantitative index comparisons of SM, RM, NTLR, and TLR on three cohorts.**

| Method | AUC | ACC | SENS | SPEC | PPV | NPV | | MCC | *F1* score |
|---|---|---|---|---|---|---|---|---|---|
| Testing set of the main cohort | | | | | | | | | |
| SM | 0.83 | 0.77 | 0.72 | **0.82** | **0.76** | 0.78 | | 0.54 | 0.74 |
| RM | 0.64 | 0.62 | 0.71 | 0.57 | 0.50 | 0.76 | | 0.27 | 0.58 |
| NTLR | 0.82 | 0.79 | 0.75 | 0.81 | 0.73 | 0.83 | | 0.56 | 0.74 |
| TLR | **0.93** | **0.84** | **0.94** | 0.77 | 0.73 | **0.95** | | **0.69** | **0.82** |
| Independent testing set 1 | | | | | | | | | |
| SM | 0.67 | 0.61 | 0.43 | 0.87 | 0.83 | 0.52 | | 0.32 | 0.57 |
| RM | 0.55 | 0.51 | 0.36 | 0.72 | 0.65 | 0.44 | | 0.08 | 0.47 |
| NTLR | 0.81 | 0.75 | 0.71 | 0.81 | 0.84 | 0.66 | | 0.51 | 0.77 |
| TLR | **0.93** | **0.86** | **0.83** | **0.89** | **0.92** | **0.78** | | **0.71** | **0.87** |
| Independent testing set 2 | | | | | | | | | |
| SM | 0.67 | 0.67 | 0.68 | 0.67 | 0.66 | 0.69 | | 0.35 | 0.67 |
| RM | 0.57 | 0.60 | 0.47 | 0.69 | 0.53 | 0.64 | | 0.16 | 0.50 |
| NTLR | 0.79 | 0.73 | 0.67 | 0.78 | 0.70 | 0.76 | | 0.46 | 0.68 |
| TLR | **0.93** | **0.84** | 0.95 | 0.75 | **0.74** | **0.96** | | **0.70** | **0.83** |
| Significance level of Delong test for methods compared with TLR | | | | | | | | | |
| Testing dataset | SM | | | RM | | | NTLR | | |
| Main cohort | 0.0025 | | | <0.0001 | | | 0.0012 | | |
| Independent set 1 | <0.0001 | | | <0.0001 | | | <0.0001 | | |
| Independent set 2 | <0.0001 | | | <0.0001 | | | <0.0001 | | |

The bold values represent the best value of an index in the comparative experiments.

need a means to compensate for the lack of lymph node ultrasound evaluation, improve the accuracy of preoperative prediction of cervical LNM, especially in the central region, assist in the development of surgical procedures, reduce the damage of the recurrent laryngeal nerve, blood vessels, and lymphatic vessels in LND.

A number of studies have shown that certain features of ultrasound examination have strong correlation with LNM, but the conclusions of these studies were inconsistent and the accuracy of LNM prediction was not high. The related research on LNM evaluation for PTC was divided into two categories: statistical methods based on clinical experience and computer-aided diagnosis method.

Some univariate and multivariate analyses have shown that ultrasound features including tumor size, thyroid invasion, and microcalcifications are independent indicators of LNM of PTC ($P < 0.05$)[9,15,16,24–26]. Wu et al. used Doppler blood flow imaging to evaluate LNM, and concluded that tumor size, blood flow, and occurrence of Hashimoto were independent factors for LNM in PTC patients ($P$ values were 0.004, 0.118, and 0.016, respectively)[17]. Nie et al. compared ultrasound images with computed tomography images of LNM in PTC patients. They found that tumor size, tumor invasiveness, and tumor location were significantly associated with LNM in the univariate analysis ($P < 0.05$)[18].

In recent years, computer-aided diagnostics, especially machine learning methods, have also been used in LNM prediction for PTC patients. Jin et al. predicted the lateral LNM by establishing a logistic regression model. Information of Hashimoto, invasiveness, multifocality, tumor number, central LNM number, TSH, and TGAb serum levels had been included into modeling, and the AUC of prediction was 0.78 in the cohort of 106 patients[20]. Liu et al. introduced the radioimics method into the LNM prediction of PTC patients. By extracting 614 features from each B-mode ultrasound image, a prediction model was established by a classical machine learning method of support vector machine (SVM). An AUC of 0.78 was obtained on the training set of 300 cases, and an AUC of 0.73 on the test set of 155 cases[22]. In addition, Liu et al. also analyzed the value of strain elastography for LNM prediction. By performing leave-one-out cross-validation on a dataset of 75 patients, it was shown that B-mode plus strain elastography provided improved LNM prediction AUC from 0.81 based on B-mode images to 0.90 based on multimodality images[21].

It should be pointed out that the above machine learning-based research was carried out under the conditions of single center, fixed machine, and fixed image acquisition protocol. In the multicenter, cross-machine, multi-operator scenario, the performance of clinical diagnostic models and traditional radiomics methods was greatly reduced, especially in the independent tests of multifocal-lesion and cross-hospital sets. The mode based on clinical information can only reach the AUC of 0.67, while the traditional radiomics can only reach the AUC of 0.56 in independent testing. This revealed several issues. First, the diagnostic model based on clinical experience is greatly influenced by observer variances. Second, the contribution of blood indicators in diagnostic models was limited, indicating that these indicators may not be the determinants of LNM prediction. Third, the high-throughput features extracted in traditional radiomics are easily affected by the imaging parameters. The performance was degraded in multiple-center scenario, indicating the traditional RM has poor generalization ability. Fourth, although studies have verified the value of ultrasound elastography for LNM prediction in small sample data, in current clinical PTC diagnosis and lymph node examination, elastography is not a widely accepted modality. The use of multimodal ultrasound in the LNM prediction of PTC patients is expected to further improve the performance of the TLR model, which is what we will continue to do in the future.

In this study, the main cohort was divided according to the factor of diagnosis time, machine, and operator. Under different data partitioning modes, more than 0.90 of AUC can be obtained in the testing set, indicating that the TLR is robust to data acquisition machines and data acquisition operator. When the data collected by a certain device or a sonographer does not appear in the training set, the TLR can also obtain a stable LNM prediction performance. There are few studies for LNM prediction in multifocal lesions PTCs because it is not possible to determine which nodules have LNM among multiple ones. In this study, we used multifocal lesions PTCs as an independent test set. If the image of a tumor was determined by the TLR model to have

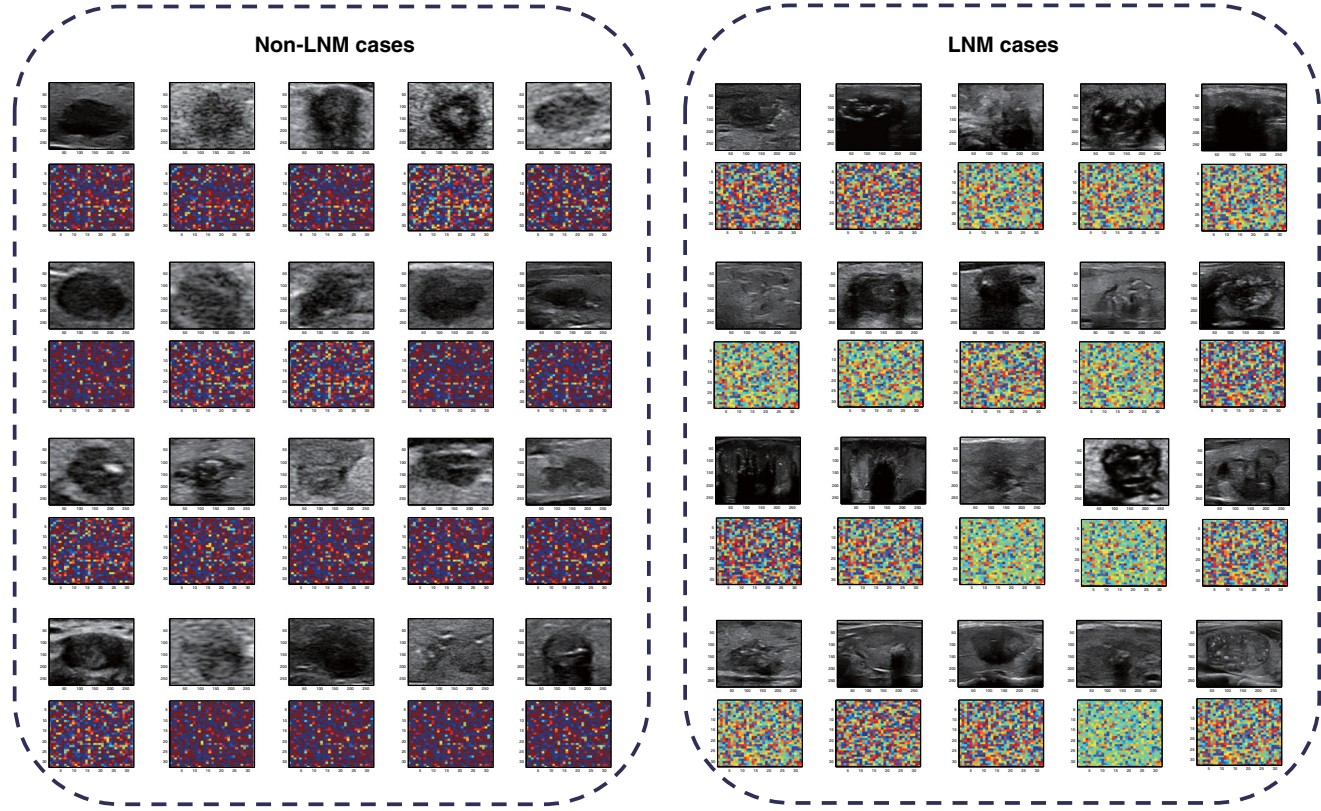

**Fig. 4 Visualization of network features of 20 cases with and without LNM, respectively.** The left side shows the network features of 20 cases without LNM, and right side shows 20 cases with LNM.

LNM, the case was LNM positive. In this scenario, the TLR also achieved an AUC of 0.93, indicating the validity of the TLR for LNM prediction of multifocal lesions PTCs. For data from another hospital that was completely different from the main cohort in terms of imaging equipment and operators, the TLR gave an AUC of 0.93 in LNM prediction. Delong test and decision curve analysis (DCA) proved that the TLR was statistically better than other methods and could bring consistent and significant benefit to PTC patients than either the treat-none or treat-all-patients scheme.

TLR gave stable high LNM prediction performance due to the following characteristics of the transfer learning[27,28]. First, when a deep learning model has shown good performance in a large dataset, the model already has very strong capabilities in image characterization. When transferring to a new task, it can avoid overfitting problem in a new training dataset, which reduces the amount of data required for modeling, and on the other hand, improves the generalization capabilities of the modeling. Second, the establishment of a new diagnostic model from a pretrained model can greatly reduce the number of training parameters and make the training process more stable and efficient. Besides, the hyperparameter optimization based on the simulated annealing algorithm largely ensures that the TLR converge to an optimal hyperparameter combination. Experiments manifested the clinical availability of the TLR model in a multicenter, cross-machine, multi-operator scenario.

The prediction results of TLR can be well incorporated into the existing PTC treatment guidelines. The 2015 American Thyroid Association management guidelines for differentiated thyroid cancer[5] recommended that for patients with PTC <4 cm without clinical evidence of LNM by clinical physical examination and radiological examination (cN0), thyroid lobectomy alone can be performed without LND; however, for patients either cytologically confirmed or highly suspicious for metastatic disease (cN1), therapeutic LND is recommended[5]. According to the 2020 National Comprehensive Cancer Network clinical practice guidelines for thyroid carcinoma[28], PTC patients with cN1 need to undergo total thyroidectomy and LND of involved compartments, while PTC patients without cervical LNM can only undergo lobectomy when tumor ≤4 cm. Consequently, the TLR model can assist in clinical decision-making for PTC patients. The prediction of LNM based on the TLR model can compensate for the limitations of preoperative clinical and ultrasound assessment of lymph nodes and allow PTC patients to receive the most reasonable treatment. For example, for PTC less than 4 cm, if cervical TNM is not detected by preoperative ultrasound but predicted by the TLR model, central LND is preferred, or at least, a second ultrasound examination, or other imaging examination should be performed to minimize the missed diagnosis of LNM.

The TRL model can also play a role in the selection of active surveillance strategies for papillary thyroid microcarcinoma (PTMC) patients. Active surveillance rather than immediate surgery of low-risk PTMC is receiving increasing attention[5]. Low-risk PTMC refers to tumors that do not have the following characteristics, including located adjacent to the trachea, possibly invading the recurrent laryngeal nerve, fine-needle biopsy findings suggesting high-grade malignancy, and presence of regional LNM[29,30]. Therefore, by predicting the state of cervical LNM, the TRL model can be used to judge whether the PTMC is suitable for active surveillance rather than surgical treatment.

## Methods

**Patients.** This study was approved by the institutional ethics committee of the involved multicenters. Three datasets were included in this study. The first two

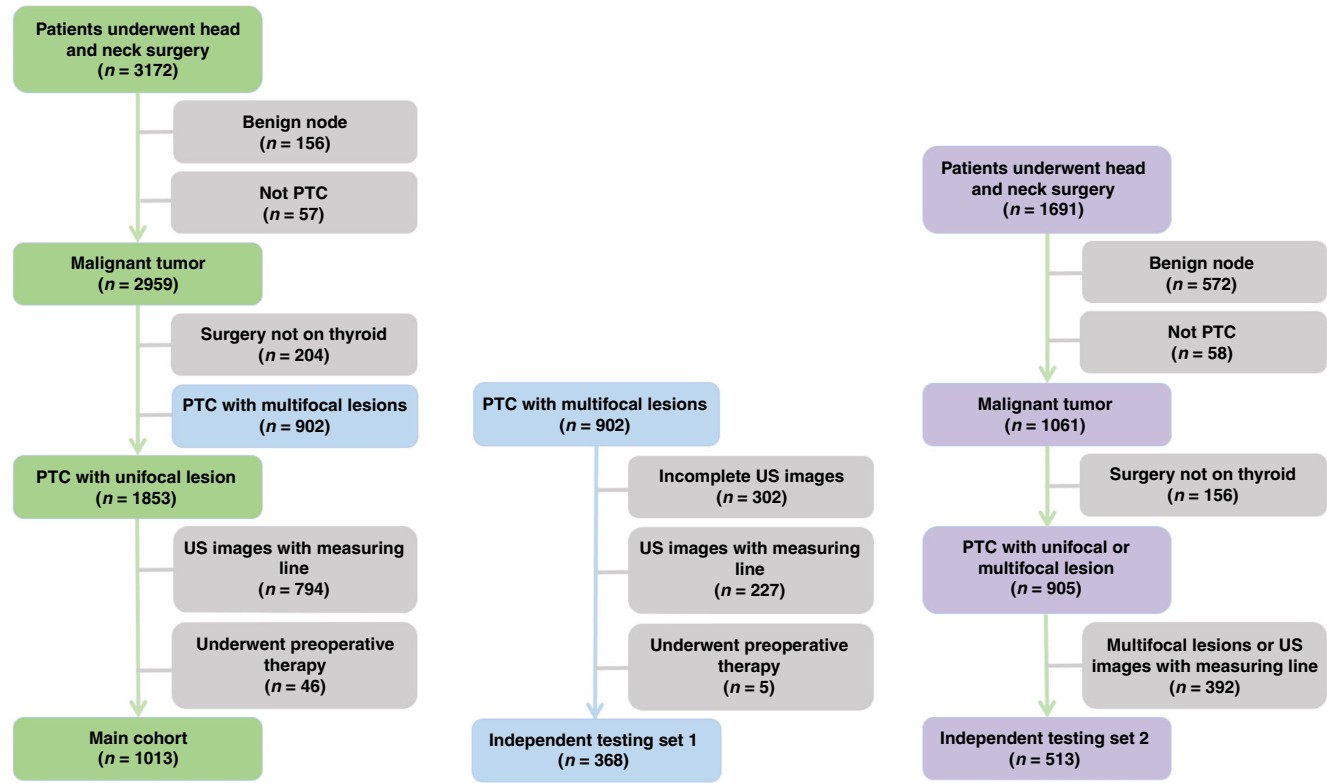

**Fig. 5 Process of the patient enrollment for main cohort and two independent testing sets.** The left side shows the patient enrollment process of the main cohort, the middle is the one of the independent testing set1, and the right side is the one of the independent testing set2.

**Table 3 Patient characteristics of three cohorts.**

| Characteristics | Main cohort | | | Independent testing set 1 | | | Independent testing set 2 | | |
|---|---|---|---|---|---|---|---|---|---|
| | LNM status | | P value | LNM status | | P value | LNM status | | P value |
| | Yes | No | | Yes | No | | Yes | No | |
| Age (mean ± SD) | 39.3 ± 11.5 | 44.8 ± 11.4 | <0.0001 | 43.1 ± 11.7 | 46.4 ± 10.5 | 0.0046 | 41.7 ± 11.9 | 45.0 ± 12.3 | 0.0415 |
| Sex | – | – | <0.0001 | – | – | 0.0022 | – | – | 0.0067 |
| Male | 142 | 145 | – | 70 | 34 | – | 75 | 68 | – |
| Female | 261 | 465 | | 147 | 117 | | 143 | 227 | |
| Tumor diameter (mm) | 15.0 ± 8.7 | 9.1 ± 5.3 | <0.0001 | 15.3 ± 9.2 | 11.5 ± 7.0 | <0.0001 | 12.6 ± 8.4 | 9.5 ± 6.0 | <0.0001 |
| Number of lesions | 1 | 1 | – | 4.1 ± 2.3 | 3.9 ± 2.0 | 0.042 | 1 | 1 | – |
| Kwak TIRADS | | | 0.027 | | | 0.9404 | | | 0.782 |
| 4A | 6.6% | 18.4% | – | 6.3% | 9.9% | – | 3.2% | 0.7% | – |
| 4B | 32.1% | 42.8% | – | 26.4% | 36.0% | – | 5.5% | 7.8% | – |
| 4C | 45.2% | 33.7% | – | 41.4% | 45.9% | – | 64.7% | 64.7% | – |
| 5 | 16.1% | 5.1% | – | 25.9% | 8.1% | – | 26.6% | 26.8% | – |
| Blood markers | | | | | | | | | |
| TSH | 2.2 ± 1.3 | 2.3 ± 2.0 | 0.6149 | 2.8 ± 5.2 | 2.3 ± 1.4 | 0.5437 | 3.0 ± 6.9 | 2.6 ± 4.0 | 0.9104 |
| TGab | 49.2 ± 151.9 | 59.1 ± 162.2 | 0.1187 | 106.5 ± 255.9 | 47.9 ± 151.5 | 0.0074 | 63.4 ± 172.0 | 101.1 ± 217.6 | 0.5553 |
| TG | 35.6 ± 68.0 | 28.20 ± 66.22 | <0.0001 | 51.4 ± 94.4 | 30.9 ± 67.7 | 0.0274 | 36.4 ± 77.1 | 16.6 ± 28.8 | 0.0912 |
| TPOAB | 62.5 ± 193.9 | 59.2 ± 179.5 | 0.4127 | 61.2 ± 168.0 | 74.8 ± 205.1 | 0.644 | 56.7 ± 165.9 | 97.6 ± 236.0 | 0.5989 |
| Total | 403 | 610 | | 217 | 151 | | 218 | 295 | |
| | 1013 | | | 368 | | 513 | | | |

datasets were selected from 3172 patients who underwent thyroid examination from April 2015 to December 2017. Among 3172 patients, 1013 PTC cases with unifocal lesions were used as the main cohort. The second dataset contained 368 PTC cases with multifocal lesions, which referred as the independent testing set 1. The third dataset was from 1691 patients at two separate hospitals from October 2008 to September 2018. A total of 513 PTC cases of unifocal lesion were selected and referred as the independent testing set 2. The data screening process for the three datasets was illustrated in Fig. 5. Verbal informed consent was obtained from all patients. The two corresponding authors of this article were authorized by the ethics committee of Fudan University Shanghai Cancer Center and Ruijin Hospital

Affiliated to Shanghai Jiaotong University to supervise the informed consent. The patient was informed at the time of ultrasound examination that the ultrasound data might be used for research purposes. After obtaining the informed consent of the patient, the patient's identity information will be recorded in the "included patients" list.

The inclusion criteria were: (1) patients underwent thyroid ultrasound diagnosis and had clear B-mode ultrasound images, (2) patients confirmed to be PTC after thyroidectomy, and (3) patients received LND and the ground truth of LNM was according to the pathologic evaluation. The exclusion criteria were: (1) there were measuring lines on the ultrasound images, (2) the nodules were too large to obtain

images covering the complete outline of the nodules even by adjusting the position of the scanning section, (3) patients received preoperative treatment, and (4) missing clinical information, such as incomplete ultrasound images for multifocal lesions and incomplete lymph node information. The patient demographics, the size, number, and the Kwak TIRADS classification of the tumor, as well as the blood markers of the main cohort, independent testing set 1, and independent testing set 2 were summarized in Table 3. For multifocal-lesion cases in independent testing set 1, every malignant nodule was confirmed by pathological diagnosis after surgery.

The main cohort was used for the model establishment and the study of the influence of various factors (diagnostic time, equipment, and operators) on the modeling performance. Fifteen machines and twenty-two sonographers were involved in data collection of the main cohort. The information of 15 machines and 22 sonographers were summarized in Supplementary Tables 5 and 6, respectively. The two independent testing sets were used to evaluate the proposed LNM prediction model.

**Method overview**. Each ultrasound radiologist involved in the acquisition of ultrasound images had more than 5 years of experience in thyroid ultrasound. Before collecting ultrasound data, all ultrasound radiologists underwent rigorous training to standardize the imaging parameter adjustment method and the ultrasound scanning procedure of the thyroid and cervical lymph nodes according to the AIUM practice guideline for performing thyroid ultrasound[31]. It is routinely required to acquire images of the longitudinal and transverse sections of the target nodules for subsequent analysis. For the large nodule beyond the display range of the probe, the image covering the complete outline of the smaller part of the nodule can be obtained by adjusting the position of the scanning section. All the data of each subcenter were gathered and reviewed by two senior ultrasound radiologists, and only the data that passed the quality control examination were included.

The main cohort was divided according to the three factors: diagnosis time, ultrasound devices, and sonographers. For the diagnosis time factor, the data of 80% patients who were diagnosed earlier was used as the cross-validation set and the last 20% as the testing set. For the ultrasound device factor, the data of three types of ultrasound machines, which collected the most data, were used as three testing sets in turn, and the data of other 14 machines were used as the

**Table 4 Hyperparameter need to be optimized in model training.**

| Index | Hyperparameter | Option |
|---|---|---|
| 1 | Neurons for top layers | 128, 256, 512, 1024 |
| 2 | Number of top layers | range from 2 to 8 |
| 3 | Neurons for the layer before the output | 32, 64 |
| 4 | Dropout rate | range from 0 to 0.5 |
| 5 | Batch size | 8, 16, 32 |
| 6 | Epochs | 50, 100, 150 |
| 7 | Activation function in top layers | softplus, relu, tanh, sigmoid, linear, elu, softmax |
| 8 | Activation function for the output layer | sigmoid, softmax |
| 9 | Kernel initializer | uniform, normal |
| 10 | Optimizer | "SGD," "RMSprop," "Adadelta," "Adam," "Adamax," "Nadam" |

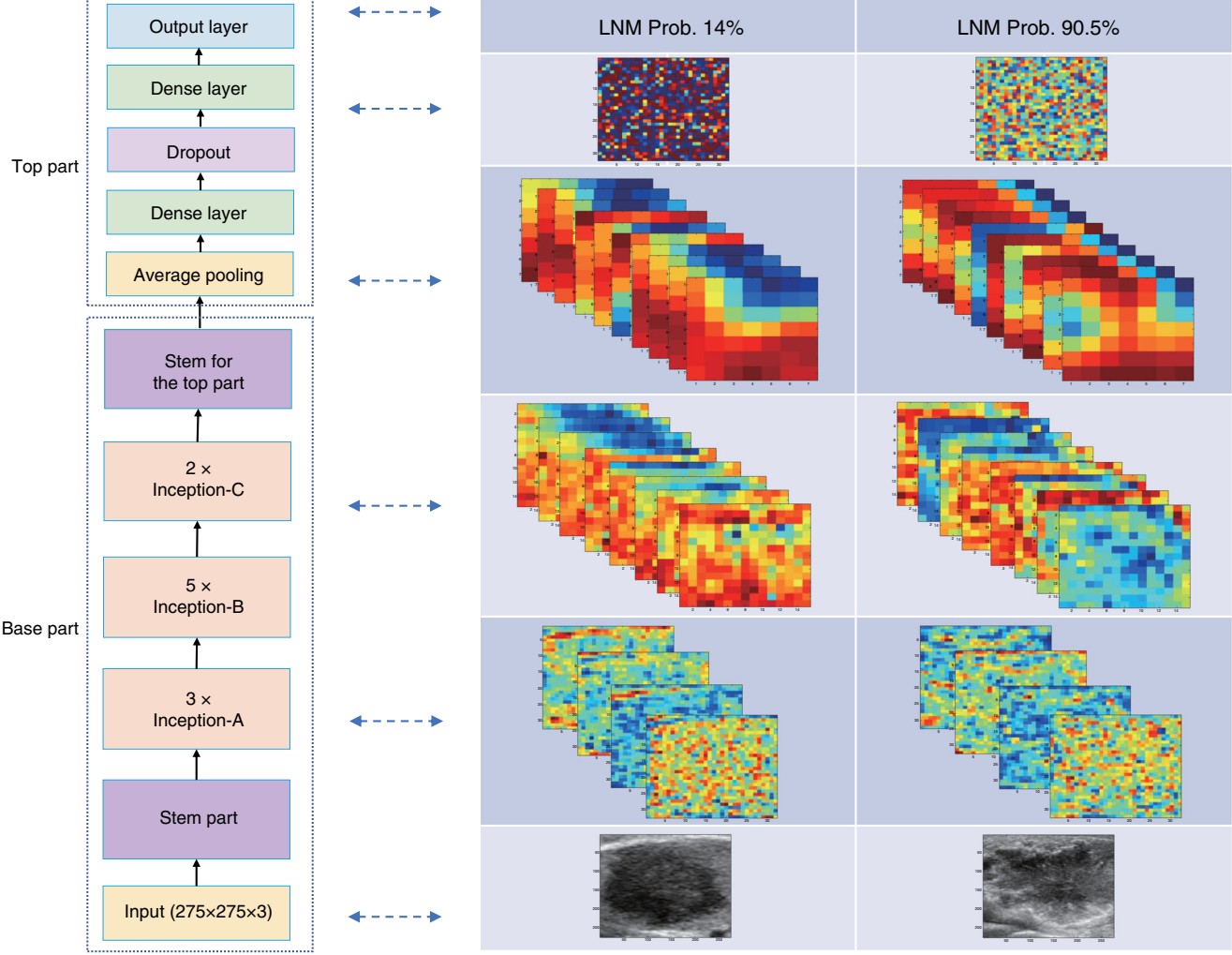

**Fig. 6 Structure of transfer learning model and illustrations of middle layer output of an LNM positive case and a negative case.** The left side shows the structure of our model, and the right side shows the middel layer output of two cases.

corresponding training set. For the sonographer factor, the data collected by three sonographers who contributed the most in the main cohort collection were used as three testing sets in turn, and the data collected by other 21 sonographers as corresponding training sets. By using different cohort partitioning methods, we aim to test whether the TLR is robust to different factors in clinical LNM assessment.

When a TLR model was established on the main cohort, we further validated the model on the two independent testing sets.

**Transfer learning**. Transfer learning is to improve learning process in new tasks by transferring knowledge from related tasks that have already been learned[32–34]. In recent years, transfer learning has been gradually applied to many fields of medical image analysis, such as image segmentation, lesion localization, and lesion pattern recognition[35,36].

The transfer learning framework used in this paper was the Inception V3 model, which was well trained with the ImageNet data to realize the natural object recognition. Figure 6 illustrated the proposed transfer learning model, the base part of which was the Inception V3 and the top part was with the structure of five layers, which is determined based on a comprehensive hyperparameter optimization process. The parameters of the transfer network were further fine-tuned during the training phase of our study. Ultrasound sonographers with more than 5 years of experience defined the region of interests (ROIs) as the input to TLR model. The ROI is a rectangle that covers a tumor lesion and 5% larger than the lesion.

The hyperparametric optimization is the most important part of the training and determines the performance of the model. The hyperparameters included the configuration parameters of the top part and the optimizing parameters of the whole model. The hyperparameters that need to be optimized were summarized in Table 4. The simulated annealing was used as the hyperparametric optimization algorithm.

The principle of simulated annealing is based on the similarity between the annealing process of solid matter and the general combinatorial optimization problem[37,38]. The simulated annealing algorithm generally starts at a higher temperature $T$, which corresponds to a larger parameter optimization space. The algorithm repeats the following steps for the solution space determined by the current hyperparameters: generating new solutions, calculating the difference among solutions, accepting or discarding the iteration, and gradually decay the temperature $T$ value. As the temperature gradually decreases, the hyperparameter space gradually shrinks. The current solution at the end of the algorithm is the approximate optimal solution. The hyperparameter space of the TLR model shown in Table 4 contains ten sets of hyperparameters. Among them, the values of hyperparameters with indexes of 1–6 are ordered, we call them ordered hyperparameters; the values of hyperparameters 7–10 are discrete and unordered, called disorder hyperparameters. In hyperparameter optimization based on simulated annealing, we adopted the strategy of optimizing ordered hyperparameters first and then optimizing disordered hyperparameters. The specific operation process is as follows:

*Step 1*: Based on large $T$ (as large scale in hyperparameter space), first select 30 sets of hyperparameter combinations. For all the hyperparameters, the value selection is performed by random search in their corresponding value ranges. For each set of the hyperparameter combination, the deep model is established and the model performance is evaluated based on the results from the cross-validation. By collecting all the results of the 30 sets of hyperparameter combinations, the initial step is completed.

*Step 2*: Start to reduce the $T$ (as relatively small scale in hyperparameter space) by narrowing down the value ranges for the ordered hyperparameters relatively near the selected values with the better results among the 30 results of the previous model evaluation. The disordered hyperparameters maintain their original options as their value range. The reduction of the updated hyperparameter space compared with the previous hyperparameter space is 50%. Based on the updated value ranges for all the hyperparameters, 30 sets of hyperparameter combinations are selected by random search and the corresponding deep model are established and evaluated. Then, this step is repeated until the determination of the values for the ordered hyperparameters.

*Step 3*: Once the values for the ordered hyperparameters are determined, the optimization for the disordered hyperparameters is further performed by grid search. The total number of the disordered hyperparameter combinations is 168 $(7 \times 2 \times 2 \times 6)$. All the 168 hyperparameter combinations are utilized for the model establishment. The selection of these hyperparameters are determined based on the best one among all the results.

*Step 4*: With the determination of all the hyperparameters by the above optimization process, the deep model can be optimally established and applied for the following testing on all the independent testing datasets.

It should be pointed out that, we intentionally made the learning rate manually optimized. The reason why the learning rate is not included in the hyperparameter optimization based on simulated annealing is that the learning rate has a wide range of values, which can be generally from $10^{-6}$ to 1. If it is put in, it will cause the hyperparameter space to expand enormously, resulting in inefficient hyperparameter searching. In addition, the adjustment of learning rate can be followed by some general principles. When the model is underfitting, appropriately increasing the learning rate can accelerate the model convergence; when the model tends to be overfitting, reducing the learning rate will benefit the modeling process.

The input image size of the base part is determined as $275 \times 275 \times 3$. After the comprehensive hyperparameter optimization process, the structure of the top part can be thus determined. The top part consists of five layers. The first layer is an average pooling layer that connects the output layer of the base part. The second layer is a dense layer for which the neuron number is 1024 and the activation function is selected as tanh. After the dense layer, a dropout layer is applied and the corresponding dropout rate is 0.12. The fourth layer is another dense layer for which the neuron number is 64 and the activation function is tanh. The fifth layer is the output layer that is a dense layer with only one neuron. The sigmoid function is the selected activation function for the output layer and the kernel initializer is selected as normal. The Adadelta method is applied as the model optimizer. The used batch size is 16 and the used epoch number is 50 in our study. The binary cross-entropy is the loss function used in our study. The applied value of learning rate is 1.0 with the selected optimizer as Adadelta.

**Method evaluation and comparison**. In order to comprehensively evaluate the TLR model under multicenter scenario, we compared the TLR model with other three methods: (1) SM based on clinical features, (2) traditional RM, and (3) non-TLR model (NTLR).

According to previous study, the clinical features included in this study were gender, age, tumor size, microcalcifications, Hashimoto's disease, TSH, TGAb, thyroglobulin (TG), and thyroid peroxidase antibody (TPOAB). The Lasso regression model for LNM prediction was attached in Supplementary Note 1.

Traditional radiomics refers to the extraction of high-throughput features from images, followed by feature selection and traditional machine learning such as SVM, adaboost, etc. to establish diagnostic models. According to three wildly accepted guidelines, including American Association Clinical Endocrinologists, American College of Endocrinology, and Associazione Medici Endocrinologi, 614 high-throughput features consisting of ten categories (demography information, size, shape, margin and boundary, orientation, position, echo pattern, posterior acoustic pattern, calcification, wavelet) have been calculated for each thyroid lesion. The details of the traditional RM[22] has been compared in this paper was described in Supplementary Note 2.

The nontransfer learning model refers to directly training the deep learning model with ultrasound image data. In this paper, the Inception V3 model is also used as the framework for nontransfer learning.

Furthermore, in order to evaluate the performance of adopted Inception V3 models, the propose TLR model is also compared with other deep learning models including VGG, ResNet, and Inception ResNet.

**Statistical analysis**. The LNM prediction results were validated by quantitative indexes including accuracy (ACC), sensitivity (SENS), specificity (SPEC), positive predictive value (PPV), negative predictive value (NPV), Matthew's correlation coefficient (MCC), and $F1$ score, which are described in Supplementary Note 3.

The Chi-squared test was used to determine whether there was any statistical difference in patient characteristics. The ROC curve was used to illustrate the overall performances of different modeling methods. Delong's test was used to test whether there is a statistical difference in LNM prediction between TLR and other methods. The DCA was used to test the clinical usefulness of the TLR model in LNM prediction.

The statistical analysis was performed by using R language (version 3.4.0, http://www.Rproject.org), IBM SPSS statistics 20.0 software (SPSS, Chicago, IL, USA), and Matlab 2017 (MathWorks, USA).

**Reporting summary**. Further information on research design is available in the Nature Research Reporting Summary linked to this article.

## Data availability
The data that support the findings of this study are available at the web repository of "https://pan.baidu.com/s/1Xaf6diOqnJ6KzM5-hlMADA" and its extraction code can be obtained from the corresponding author upon a separate request.

## Code availability
The codes of the proposed method are also available at the web repository of "https://pan.baidu.com/s/10Gm7JAZ0E3DcWpU08Hqzqw" with the extraction code as 8n49.

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

## Acknowledgements

This work was supported by the National Natural Science Foundation of China (91959127 and 81830058).

## Author contributions

J.h.Y. proposed and designed the study and all the methods, and wrote the manuscript. Y.h.D. proposed and realized the method. T.t.L. arranged the data and realized the methods for comparison. J.h.Y., Y.h.D., and T.t.L. carried out the experiments and analyzed the data. J.Z., X.h.J., T.l.X., S.c.Z., J.w.L., and Y.G. collected the data and performed the research. Y.y.W., J.q.Z., and C.C. provided supports for the research, and designed the method. All authors supplied comments and revised the paper.

## Competing interests

The authors declare no competing interests.
