## [Peer Review File · Nature Communications]

Reviewers' Comments:

Reviewer #1:

Remarks to the Author:

As a clinical professor, my review is limited to the clinical applications of this work.

The authors use a dataset of ultrasound images associated with pathologically confirmed papillary thyroid carcinoma (PTC) patients to predict lymph node metastasis. Using this dataset a Transfer Learning Radiomics (TLR) model is developed and its performance is evaluated in relation to several parameters. The TLR model performance is compared with three other methods including a statistical model based on clinical information, traditional radiomics model and non-transfer learning radiomics model. The methods used perform their analysis based on sonographers' delineation of a Region of Interest (ROI) around the primary thyroid lesion. The outcome of this analysis is to determine an overall likelihood of lymph node metastasis. The analysis does not localize the metastasis to any given lymph node or region.

Major Comments

I have two major comments:

1) In current clinical practice, ultrasound evaluation of the cervical lymph nodes is applied not only to identify whether lymph node involvement exists, but also to localize metastasis to specific cervical lymph node levels. The neck has a well-developed lymphatic network, and extensive removal of cervical lymph nodes through radical neck dissection is associated with significant morbidity, and is rarely performed in Western countries [1]. Selective neck dissection, in which only cervical lymph node levels that are identified to contain biopsy-confirmed tumor involvement are resected, is currently the method of choice. This compartment-based approach requires that the involved lymph nodes be specifically localized. In the method outlined in this study, the authors are able to successfully predict the risk of lymph node involvement in patients, but do not localize metastasis to specific lymph node levels. Identifying lymph node involvement risk without knowledge of its localization may not be sufficient in itself to significantly impact surgical management of PTC patients. Can the authors explain how they believe their method can guide clinical decision-making, given that it does not localize lymphatic involvement?

2) The paper should be revised to elaborate on the clinical relevance of this method by stating how the outcome of TLR can be incorporated into the existing clinical best practices, such as those recommended by the National Comprehensive Cancer Network (NCCN) for treatment of PTC.

References

1. Sakorafas, George H., Dimitrios Sampanis, and Michael Safioleas. "Cervical lymph node dissection in papillary thyroid cancer: current trends, persisting controversies, and unclarified uncertainties." *Surgical oncology* 19.2 (2010): e57-e70.

Reviewer #2:

Remarks to the Author:

The manuscript develops a transfer learning radiomics model for preoperative prediction of lymph node metastasis. It is interesting and has some clinical implications. The results have shown good performance of the proposed method. But there are still some problems in the manuscript as described below:

1. Since this was a multicenter, cross-machine, multi-operator prospective study, strict quality controls should be taken throughout the entire procedure. For example, how to obtain the best images, which section were chosen (axial images or sagittal images). etc.
2. The inclusion criteria and exclusion criteria didn't mention the nodule size. Actually, the image

couldn't contain all information if the nodule was too large to be covered by the probe.

3. In table 1, what is definition of Tumor Grade? Pathology or TIRADS? If the latter, please indicate which guidelines was cited.

4. The network structure shown in Figure 2 seems to be wrong. As shown in Figure 2, Base part network is the transfer Inception V3 model and the output of softmax is a category result. It seems that it is impossible to train and get the prediction model by using the softmax result as the input of proposed Top part network. In addition, are the transfer network parameters further fine turned?

5. In this manuscript, simulated annealing method is used for hyperparametric optimization. However, the authors did not give the specific hyperparametric optimization steps with simulated annealing. How to use optimization algorithm to update the hyper parameters listed in this manuscript? Moreover, the authors need to briefly introduce the principle of simulated annealing method even if it is not an innovative content.

6. In independent testing set 1, for multifocal-lesion cases, the authors interpreted that if the image of one nodule was determined by the TLR model to have LNM, the case was classified as LNM positive. Was every nodule confirmed malignance by pre-operation FNA or CNB or surgery? If not, this would bias the final result.

7. In addition, there is no comparative experiment to verify the effectiveness of simulated hyperparametric optimization. Is it really useful or just theoretical? At the same time, some important hyper parameters are not listed and optimized, such as learning rate.

8. The proposed method needs to be compared with other existing deep learning methods, such as VGG, ResNet, Inception ResNet and so on.

Author's Response to the Reviewers' Comments

We would like to express our appreciation for all these valuable and helpful comments and suggestions which guided us for improving this paper. Following the suggestions, we revised the manuscript carefully. All the revisions have been marked in red in the revised manuscript. We are now responding to the comments one by one as follows.

Reviewer's Comments

=====

Reviewer: 1

Comments to the Author

As a clinical professor, my review is limited to the clinical applications of this work.

The authors use a dataset of ultrasound images associated with pathologically confirmed papillary thyroid carcinoma (PTC) patients to predict lymph node metastasis. Using this dataset a Transfer Learning Radiomics (TLR) model is developed and its performance is evaluated in relation to several parameters. The TLR model performance is compared with three other methods including a statistical model based on clinical information, traditional radiomics model and non-transfer learning radiomics model. The methods used perform their analysis based on sonographers' delineation of a Region of Interest (ROI) around the primary thyroid lesion. The outcome of this analysis is to determine an overall likelihood of lymph node metastasis. The analysis does not localize the metastasis to any given lymph node or region.

Major Comments

I have two major comments:

1. In current clinical practice, ultrasound evaluation of the cervical lymph nodes is applied not only to identify whether lymph node involvement exists, but also to localize metastasis to specific cervical lymph node levels. The neck has a well-developed lymphatic network, and extensive removal of cervical lymph nodes through radical neck dissection is associated with significant morbidity, and is rarely performed in Western countries [1]. Selective neck dissection, in which only cervical lymph node levels that are identified to contain biopsy-confirmed tumor involvement are resected, is currently the method of choice. This compartment-based approach requires that the involved lymph nodes be specifically localized. In the method outlined in this study, the authors are able to successfully predict the risk of lymph node involvement in patients, but do not localize metastasis to specific lymph node levels. Identifying lymph node involvement risk without knowledge of its localization may not be sufficient in itself to significantly impact surgical management of PTC patients. Can the authors explain how they believe their method can guide clinical decision-making, given that it does not localize lymphatic involvement?

Author Response: Thank you for your valuable comments. Answering your question can reflect the value of our work more clearly. Accurately predicting the risk of PTC lymph node metastasis (LNM) can bring the following three aspects to clinical decision-making.

1. The 2015 American Thyroid Association (ATA) management guidelines [5] for differentiated thyroid cancer recommended that for patients with PTC <4 cm without clinical evidence of LNM by clinical physical examination and radiological examination (cN0), thyroid lobectomy alone can be performed without lymph node dissection (LND). However, for patients with apparent LNM that are either cytologically confirmed or highly suspicious for metastatic disease (cN1), therapeutic LND are recommended. According to the 2020 National Comprehensive Cancer Network (NCCN) clinical practice guidelines for thyroid carcinoma, PTC patients with cN1 need to undergo total thyroidectomy and LND of involved compartments, while PTC patients without cervical LNM can only undergo total thyroidectomy or lobectomy. According to these two widely accepted guidelines and your comments, selective resection of lymph nodes that are highly suspected of metastasis is indeed the current clinical practice. However, the diagnostic guidelines encounter many dilemmas in practice. How to determine whether a lymph node is suspected of metastasis itself is not easy, especially for central lymph nodes. Preoperative ultrasound can only detect 20%-31% of central cervical LNM, and may only change the surgical procedure of 20% patients [11]-[13]. In addition, the ultrasound diagnosis of LNM is highly dependent on the experience of sonographers. Consequently, the TLR model can assist in clinical decision-making for PTC patients. The prediction of LNM based on the TLR model can compensate for the limitations of preoperative clinical and ultrasound assessment of lymph nodes and allow PTC patients to receive the most reasonable treatment. For example, for PTC less than 4cm, if cervical LNM is not detected by preoperative ultrasound but predicted as high LNM possibility by the TLR model, central LND is preferred, or at least, a second ultrasound examination, or other imaging examination should be performed to minimize the missed diagnosis of LNM.
2. Clinically, for patients with PTC, LNM usually occurs first in the central region, followed by the lateral region due to the direction of lymphatic drainage. At present, ultrasound can diagnose lateral neck LNM with relatively high accuracy. The diagnosis accuracy, however, still depends on the experiences of sonographer. Furthermore, the diagnosis rate of central LNM is low given the anatomic structure of the central region, which cannot be visualized well by ultrasound, such as the posterior tracheal area, posterior esophageal area, posterior pharyngeal area, and mediastinal area [11]-[13]. Therefore, we can reasonably doubt that some patients with LNM can only be diagnosed when the central LNM develops to the lateral cervical lymph node. The proposed TLR model can predict the LNM of PTC patients with reasonably high accuracy, and the prediction covers central LNM, central LNM combined with lateral LNM, and lateral LNM. In this sense, TLR model can not only make up for the misdiagnosis of LNM caused by the lack of experience of sonographers, but also improve the diagnosis efficiency of central LNM, therefore provides an important reference for doctors to choose treatment options.
3. TLR model may be used in clinical decision-making for low-risk papillary thyroid microcarcinoma (PTMC: PTC \leq 1cm) patients. There is a growing opinions support the strategy of active monitoring rather than surgical treatment for low-risk PTMC, that is, PTMC without clinically evident metastases or local invasion, and no convincing cytologic evidence of aggressive disease [5]. We believe that the TRL model can help to judge whether PTMC is suitable for active monitoring rather than surgical treatment by predicting whether cervical lymph nodes have metastasis or not, combined

with other characteristics of PTMC, which is of great significance to reduce the overtreatment of PTMC.

Corrections have been made in the revised version. Please see page 3, line 48; page 3, line 57-59; page 21, line 382-389; page 25-26, line 466-490.

2) The paper should be revised to elaborate on the clinical relevance of this method by stating how the outcome of TLR can be incorporated into the existing clinical best practices, such as those recommended by the National Comprehensive Cancer Network (NCCN) for treatment of PTC.

Answer: Thank you for your comment and suggestion. As you suggested, the outcome of TLR can be incorporated into the thyroid diagnosis and treatment guidelines such as the 2020 NCCN guidelines and the 2015 ATA guidelines.

First of all, for the patients with PTC less than 1cm, i.e. PTMC, TLR model is of good value in the selection of active monitoring strategy. For PTMC patients with low-risk of LNM diagnosed by TLR model, as long as the tumor is not located adjacent to the trachea and does not invade the recurrent laryngeal nerve, this part of patients is more suitable for active monitoring rather than surgical treatment.

No matter in the 2015 American Thyroid Association (ATA) management guidelines or the 2020 National Comprehensive Cancer Network (NCCN) clinical practice guidelines, the basis for choosing therapeutic LND is whether the patient is cN1, that is to say, the patient is cytologically confirmed or highly critical for metastatic disease. As we answered in the previous question, ultrasound diagnosis of LNM is affected by the subjective experience of sonographers on the one hand. And on the other hand, due to the thyroid structure itself, the accuracy of ultrasound diagnosis of central LNM is very low. Therefore, TLR provides a new method for the clinical diagnosis of cN1.

Corrections have been made in the revised version. Please see page 25-26, line 466-490.

Reviewer #2 (Remarks to the Author):

The manuscript develops a transfer learning radionics model for preoperative prediction of lymph node metastasis. It is interesting and has some clinical implications. The results have shown good performance of the proposed method. But there are still some problems in the manuscript as described below:

1. Since this was a multicenter, cross-machine, multi-operator prospective study, strict quality controls should be taken throughout the entire procedure. For example, how to obtain the best images, which section were chosen (axial images or sagittal images). etc.

Answer: Thank you for your comment. Before collecting ultrasound data, all ultrasound radiologists involved in the acquisition of ultrasound images have underwent rigorous training to standardize the imaging adjustment method and the ultrasound scanning procedure of the thyroid and cervical lymph nodes according to the AIUM practice guideline for performing thyroid ultrasound. Each ultrasound radiologist involved in the acquisition of ultrasound images had more than 5 years of experience in thyroid ultrasound. Both longitudinal and transverse sections of the target nodules were acquired for analysis. All the data of each sub-center were gathered and reviewed by two senior ultrasound radiologists, and only the data that passed the quality control examination were included. Corresponding

descriptions have been supplemented in the revised version. Please see page 7, line 133-143.

2. The inclusion criteria and exclusion criteria didn't mention the nodule size. Actually, the image couldn't contain all information if the nodule was too large to be covered by the probe.

Answer: For nodules with a maximum diameter greater than the imaging field of the probe, a full view of the smaller part of the nodule can be obtained by adjusting the position of the scanning section. However, if the images covering the complete outline of the nodules cannot be obtained by adjusting the ultrasound probe position, such large nodules would be excluded from the study.

Corresponding descriptions have been supplemented in the revised version. Please see page 6, line 107-109.

3. In table 1, what is definition of Tumor Grade? Pathology or TIRADS? If the latter, please indicate which guidelines was cited.

Answer: Thank you for your comment. The information provided in Table 1 was the Kwak TIRADS classification of the nodules rather than tumor grade.

Corrections have been made and the guideline has been cited in the revised version. Please see page 6, line 111-112; Table 1; Ref 23.

4. The network structure shown in Figure 2 seems to be wrong. As shown in Figure 2, Base part network is the transfer Inception V3 model and the output of softmax is a category result. It seems that it is impossible to train and get the prediction model by using the softmax result as the input of proposed Top part network. In addition, are the transfer network parameters further fine turned?

Answer: Thanks very much for pointing out this mistake. Indeed the softmax layer including the fully-connected layer at the top of the original Inception V3 model was not included in our proposed model. Apologize for the wrong information here. Since the structure of the base part as Inception V3 is not about the novelty of our study, we have revised and simplified the demonstrated figure of the applied model as the updated Figure 2. The parameters of the transfer network were further fine-tuned during the training phase of our study.

The related descriptions have been added to the section of the revised manuscript. Please see page 9, line 165-166; page 9, Figure 2.

Fig. 2. Structure of transfer learning model and illustrations of middle layer output of an LNM positive case and a negative case. (Updated)

5. In this manuscript, simulated annealing method is used for hyperparametric optimization. However, the authors did not give the specific hyperparametric optimization steps with simulated annealing. How to use optimization algorithm to update the hyper parameters listed in this manuscript? Moreover, the authors need to briefly introduce the principle of simulated annealing method even if it is not an innovative content.

Answer: By following your suggestions, we added a description of the principle of the simulated annealing algorithm in the revised manuscript. And added a subsection to describe in detail the optimization steps of hyperparametric optimization by using simulated annealing.

Simulated annealing algorithm is a general probability algorithm, which is used to find the optimal solution in a large search space. The principle of simulated annealing is based on the similarity between the annealing process of solid matter and the general combinatorial optimization problem. The algorithm is to first have a relatively high “temperature”, T . Based on the high T , the algorithm may search in a large range of the entire hyperparameter space. In this situation, if we tried 30 sets of hyperparameters, the 30 points of the hyperparameter space were distributed in a large scale. Then we may find some of the hyperparameters have better model performance compared with others. According to simulated annealing algorithm, we need to reduce the value of T now, which means to search in a relatively small range of hyperparameter space. However, the question now is, where should we perform such relatively small-range search in the hyperparameter space? The answer is that based on the previous results of the 30 sets of hyperparameters, we should further perform the small-range search relatively near the points denoted by the sets of hyperparameters that have better model performances. It can be seen that this process narrows down the further search space for hyperparameters. By repeating this process and continuously reducing the “ T ”, we may finally converge to a set of hyperparameters that may be used to establish a model with an optimal performance.

As for the optimization of the hyperparameter space in TLR model, we divide these 10 groups of hyperparameters (as shown in Table 2) into two groups for optimization. One group is hyperparameters

with index of 1-6 as shown in Table 2. Their values are ordered and we call them ordered hyperparameters. The other group is hyperparameter with index of 7-10. Their values are discrete and unordered, so they are called unordered hyperparameters. In the optimization of these 10 groups of hyperparameters, we optimize the ordered hyperparameters and the unordered hyperparameters in two steps. The specific operation steps are described in detail in the revised manuscript.

The principle of simulated annealing method was described in page 10, line 180-194. The specific hyperparametric optimization steps with simulated annealing method was added in page 10-12, line 195-226.

6. In independent testing set 1, for multifocal-lesion cases, the authors interpreted that if the image of one nodule was determined by the TLR model to have LNM, the case was classified as LNM positive. Was every nodule confirmed malignance by pre-operation FNA or CNB or surgery? If not, this would bias the final result.

Answer: In the diagnosis of multiple nodules, the surgical results were used as the ground truth. After the operation, the pathological diagnosis of each nodule was made to determine whether it was benign or malignant.

The description has been added in revised manuscript. Please see page 6, line 113-115.

7. In addition, there is no comparative experiment to verify the effectiveness of simulated hyperparametric optimization. Is it really useful or just theoretical? At the same time, some important hyper parameters are not listed and optimized, such as learning rate.

Answer: Thanks for pointing this out and also thanks for the nice suggestion of the following comment 8. We may directly see the effectiveness of the hyperparameter optimization algorithm in our study compared with the results of other deep models shown in Table 7. Based on our work and the achieved results, it is indeed and really useful. From the table listing the possible options of different hyperparameters even without learning rate, it can be seen that the total number of only hyperparameter combinations is already as high as 1,016,064. If we do not apply an algorithm for the search of the optimal hyperparameter set and just do the grid search which is to attempt all of them, the time cost is tremendous and definitely unendurable. Therefore generally, the hyperparameter optimization of the deep model is performed by experienced researchers. However the optimal solution is not guaranteed. In our study, the hyperparameter optimization algorithm is indeed useful, which is one of the reasons that the work achieved a good result.

There are two reasons why we chose to carry out the learning rate separately instead of optimizing it in simulated annealing. One is that the super parameter space composed of 10 groups of parameters shown in Table 2 is already very large. As you may see, the total number of hyperparameter combinations is already as high as 1,016,064 even without the learning rate. The range of learning rate is very wide, usually from 10^{-6} to 1. If the learning rate is also put into simulated annealing to optimize, the space of super parameters will be expanded to the range of difficult to effectively calculate. The second reason is that, unlike the hyperparameters in Table 2, the regulation of learning rate has principles to follow. When the model is under-fitting, appropriately increasing the learning rate can accelerate the convergence rate; when the model is over-fitting, reducing the learning rate may benefit the modeling process.

Corresponding descriptions have been added in revised manuscript. Please see page 11-12, line 218-226.

8. The proposed method needs to be compared with other existing deep learning methods, such as VGG, ResNet, Inception ResNet and so on.

Answer: By following your suggestions, we compared our TLR model with VGG, ResNet, and Inception ResNet in the three cohorts. We have added the performance comparison in the revised manuscript, also as shown as following. Please see page 20, Table 7.

Method	AUC	ACC	SENS	SPEC	PPV	NPV	MCC	F1score
Testing set of the main cohort								
VGG	0.77	0.74	0.65	0.80	0.69	0.78	0.46	0.67
ResNet	0.77	0.71	0.47	0.87	0.70	0.71	0.37	0.56
InceptionResNet	0.75	0.71	0.46	0.89	0.73	0.71	0.39	0.56
Our model	0.93	0.84	0.94	0.77	0.73	0.95	0.69	0.82
Independent testing set 1								
VGG	0.58	0.55	0.33	0.86	0.77	0.47	0.22	0.46
ResNet	0.58	0.55	0.34	0.85	0.77	0.47	0.22	0.47
InceptionResNet	0.56	0.56	0.45	0.71	0.69	0.47	0.16	0.55
Our model	0.93	0.86	0.83	0.89	0.92	0.78	0.71	0.87
Independent testing set 2								
VGG	0.66	0.65	0.59	0.70	0.59	0.70	0.29	0.59
ResNet	0.59	0.61	0.55	0.65	0.54	0.66	0.20	0.54
InceptionResNet	0.64	0.63	0.43	0.77	0.58	0.65	0.21	0.49
Our model	0.93	0.84	0.95	0.75	0.74	0.96	0.70	0.83

We appreciate for Editors/Reviewers' nice work earnestly, and hope that the correction will meet the quality requirement of NC. Once again, thank you very much for your comments and suggestions.

Sincerely,
Authors

Reviewers' Comments:

Reviewer #1:

Remarks to the Author:

Thank you for your responses. My comments are appropriately answered.

Reviewer #2:

Remarks to the Author:

The authors have improved a lot on the previous version and revised some of the previous problems. But there remain a few problems to be solved. So a minor revision is required. The problem is as follows:

- 1.The proposed method applied simulated annealing for optimizing the hyperparametric of the network. Is the proposed network more effective after the hyperparametric optimization?
- 2.An ablation experiment is needed as a comparison study to verify its effectiveness.
- 3.In addition, the time cost needs to be discussed.

Author's Response to the Reviewers' Comments

We would like to express our appreciation for all these valuable and helpful comments and suggestions which guided us for improving this paper. Following the suggestions, we revised the manuscript carefully. All the revisions have been marked in red in the revised manuscript. We are now responding to the comments one by one as follows.

Reviewer's Comments

=====

Reviewer #1 (Remarks to the Author):

Thank you for your responses. My comments are appropriately answered.

Re: Thanks again for your comments.

Reviewer #2 (Remarks to the Author):

The authors have improved a lot on the previous version and revised some of the previous problems. But there remain a few problems to be solved. So a minor revision is required. The problem is as follows:

1. The proposed method applied simulated annealing for optimizing the hyperparametric of the network. Is the proposed network more effective after the hyperparametric optimization?

Author Response: Thank you for your approval of our last revision. The hyperparameter optimization based on the simulated annealing algorithm largely ensures that the transfer learning radiomics (TLR) converge to an optimal hyperparameter combination. This is almost impossible to achieve by manual adjustment. According to your next suggestion, we have added a supplementary experiment to prove the effect of hyperparameter adjustment based on the simulated annealing algorithm.

2. An ablation experiment is needed as a comparison study to verify its effectiveness.

Author Response: Thanks for the nice suggestion. The proposed TLR model can achieve the performance in the manuscript is mainly due to two factors, one is the strategy of transfer learning, and the other is the hyperparameter optimization based on simulated annealing algorithm. Therefore, we conducted the ablation experiments on the testing set of the main cohort and the other two independent testing sets. For the convenience of description, we denote the TLR with or without the transfer learning as T+ and T-, and the TLR with or without hyperparameter optimization as H+ and H-. The following table shows the results of the ablation experiments.

Table. Results of the ablation experiments.

Transfer learning (T) and hyperparametric optimization (H)	AUC value for the comparison of results		
	Testing set of the main cohort	Independent testing set 1	Independent testing set 2
T -, H -	0.718	0.604	0.614
T +, H -	0.791	0.650	0.594
T -, H +	0.817	0.808	0.792
T +, H +	0.927	0.928	0.932

Related experimental results and explanations have been added to the revised manuscript. Please see page 20-21, Line 367-371; Page 21, Table 8; Discussion, page 26, Line 478-480.

3. In addition, the time cost needs to be discussed.

Author Response: For the time cost, the related hardware and software environment for our study has been added in the revised manuscript. Please see page 21, Line 375-380.

The used graphics card is TITAN XP with the CUDA core number as 3840 and the graphic memory as 45008 MB. For the coding of the deep model, the applied TensorFlow is the GPU version of 1.14.0 and the Keras is utilized with its 2.3.0 version. For the establishment of one deep model in our study, the model training process generally takes 6 days with the utilization of hyperparametric optimization. For the prediction on one image data, the time cost of the model inference is around 10 ms.

We appreciate for Editors/Reviewers' nice work earnestly, and hope that the correction will meet the quality requirement of NC. Once again, thank you very much for your comments and suggestions.

Sincerely,

Authors